# Pathogenesis, Clinical Features, and Treatment of Patients with Myelin Oligodendrocyte Glycoprotein (MOG) Autoantibody-Associated Disorders Focusing on Optic Neuritis with Consideration of Autoantibody-Binding Sites: A Review

**DOI:** 10.3390/ijms241713368

**Published:** 2023-08-29

**Authors:** Keiko Tanaka, Takeshi Kezuka, Hitoshi Ishikawa, Masami Tanaka, Kenji Sakimura, Manabu Abe, Meiko Kawamura

**Affiliations:** 1Department of Animal Model Development, Brain Research Institute, Niigata University, 1-757 Asahimachi-dori, Chuoku, Niigata 951-8585, Japan; 2Department of Multiple Sclerosis Therapeutics, School of Medicine, Fukushima Medical University, 1 Hikarigaoka, Fukushima 960-1247, Japan; 3Department of Ophthalmology, Tokyo Medical University, Tokyo 160-0023, Japan; 4Department of Orthoptics and Visual Science, School of Allied Health Sciences, Kitasato University, Kanagawa 252-0373, Japan; 5Kyoto MS Center, Kyoto Min-Iren Chuo Hospital, Kyoto 616-8147, Japan; 6Division of Instrumental Analysis, Center for Coordination of Research Facilities, Institute for Research Administration, Niigata University, Niigata 951-8585, Japan

**Keywords:** myelin oligodendrocyte glycoprotein, autoantibody, optic neuritis, antibody-binding epitope, animal model

## Abstract

Although there is a substantial amount of data on the clinical characteristics, diagnostic criteria, and pathogenesis of myelin oligodendrocyte glycoprotein (MOG) autoantibody-associated disease (MOGAD), there is still uncertainty regarding the MOG protein function and the pathogenicity of anti-MOG autoantibodies in this disease. It is important to note that the disease characteristics, immunopathology, and treatment response of MOGAD patients differ from those of anti-aquaporin 4 antibody-positive neuromyelitis optica spectrum disorders (NMOSDs) and multiple sclerosis (MS). The clinical phenotypes of MOGAD are varied and can include acute disseminated encephalomyelitis, transverse myelitis, cerebral cortical encephalitis, brainstem or cerebellar symptoms, and optic neuritis. The frequency of optic neuritis suggests that the optic nerve is the most vulnerable lesion in MOGAD. During the acute stage, the optic nerve shows significant swelling with severe visual symptoms, and an MRI of the optic nerve and brain lesion tends to show an edematous appearance. These features can be alleviated with early extensive immune therapy, which may suggest that the initial attack of anti-MOG autoantibodies could target the structures on the blood–brain barrier or vessel membrane before reaching MOG protein on myelin or oligodendrocytes. To understand the pathogenesis of MOGAD, proper animal models are crucial. However, anti-MOG autoantibodies isolated from patients with MOGAD do not recognize mouse MOG efficiently. Several studies have identified two MOG epitopes that exhibit strong affinity with human anti-MOG autoantibodies, particularly those isolated from patients with the optic neuritis phenotype. Nonetheless, the relations between epitopes on MOG protein remain unclear and need to be identified in the future.

## 1. Introduction

Myelin oligodendrocyte glycoprotein (MOG), which is exclusively expressed in oligodendrocytes, is a component of the outer surface of myelin in the central nervous system (CNS) [1]. Although a quantitatively minor component, MOG has strong antigenicity. In fact, MOG was initially identified as an immunodominant target for demyelinating autoantibodies in a guinea pig model of experimental autoimmune encephalomyelitis (EAE) [2,3]. Subsequent studies have demonstrated that immunization with MOG peptides can induce an EAE variant that exhibits many of the clinical and pathologic characteristics of multiple sclerosis (MS) in both rats and primates. Litzenburger T. et al. demonstrated the persistent presence of MOG-reactive B cells in the peripheral immune system and suggesting their potential roles as modifiers in inflammatory CNS diseases using transgenic mice producing MOG-specific immunoglobulins [2]. Anti-MOG autoantibodies have been detected in many EAE variants, inciting many promising studies in patients with CNS demyelinating diseases. Over the years, extensive studies conducted in patients with MS have investigated the presence of anti-MOG autoantibodies using Western blotting and enzyme-linked immunosorbent assays targeting recombinant mouse MOG, without clear relation and specificity with MS [4]. Pöllinger B. et al. developed transgenic mice bearing MOG peptide-specific T cell receptors, resulting in spontaneous relapsing–remitting EAE along with the expansion of autoreactive B cells that produce autoantibodies binding to a conformational epitope on the native MOG protein [3]. This important finding that the pathogenic autoantibodies recognize a conformational epitope on the native antigen protein led to the designation of the human anti-MOG autoantibody-associated disease [5,6]. 

In recent years, the presence of anti-MOG autoantibodies has been extensively tested in patients with CNS inflammatory diseases using a cell-based assay that preserves the conformational structure of the full-length human MOG [7]. The International Consensus Group on MOG autoantibody-associated disease (MOGAD) has proposed that the diagnostic criteria for MOGAD should include the presence of anti-MOG autoantibodies detected using cell-based assays [8]. MOGAD is typically associated with acute disseminated encephalomyelitis (ADEM), optic neuritis (ON), and transverse myelitis (TM) and is less commonly associated with cerebral cortical encephalitis, brainstem or cerebellar symptoms, and clinical presentations including the combination of several phenotypes and sometimes accompanies other autoantibodies such as anti-*N*-methy-D aspartate receptor (NMDAR) autoantibodies with symptoms of autoimmune encephalitis [9]. MOGAD can have a monophasic or relapsing disease course; therefore, detecting anti-MOG autoantibodies using cell-based assays is essential for diagnostic accuracy. 

The majority of adult patients who are positive for anti-MOG autoantibodies exhibit ON or TM, while ADEM with or without ON is the most frequent presentation in pediatric patients with MOGAD. Factors that determine age-specific clinical phenotypes and CNS lesions in patients with MOGAD remain unclear. Furthermore, the MOG protein function and the pathogenicity of anti-MOG autoantibodies in MOGAD have not been fully clarified. 

The development of appropriate models is critical to elucidate the specific functions of anti-MOG autoantibodies. However, the low affinity of human anti-MOG autoantibodies for mouse MOG has hindered the establishment of reliable models. Several studies have shown that the recognition of MOG by anti-MOG autoantibodies involves highly complex mechanisms [10,11,12]. The antigen-recognition patterns of MOGAD might differ from those of anti-aquaporin 4 (AQP4) autoantibody-related neuromyelitis optica spectrum disorders (NMOSDs) and anti-NMDAR autoantibody-related autoimmune encephalitis, two clinical presentations with well-characterized antibody-binding sites on the disease-related antigen [13,14].

In this article, we summarize the latest studies examining anti-MOG autoantibody recognition in patients with MOGAD and provide a review of the clinical characteristics of this disease with a focus on ON, the most prevalent phenotype of MOGAD. 

## 2. Comparison of Clinical Characteristics of ON between Anti-MOG Autoantibody-Positive and Anti-AQP4 Autoantibody-Positive Patients

### 2.1. Epidemiology

We have recently reported ON’s clinical and epidemiologic characteristics based on the neuroimmunological background in a large Japanese cohort of 531 patients [15]. In that study, 12% of the patients (n = 66, 84% females) were anti-AQP4 autoantibody-positive, with a median onset age of 51 years, whereas 10% of the patients (n = 54, 51% females) were anti-MOG autoantibody-positive, with a median onset age of 43 years. In that cohort, 77% (n = 440, 64% females) of the patients were negative for both anti-MOG and anti-AQP4 autoantibodies, with a median onset age of 48 years, and included patients with MS (8%) (Table 1). The anti-AQP4 autoantibody positivity increased with age whereas the anti-MOG autoantibody positivity exhibited biphasic peaks in the fourth and sixth decades of life [15].

A cross-sectional cohort study conducted in the Mayo Clinic, which included 246 patients with recurrent ON, revealed that 19% and 13% of the patients were positive for anti-AQP4 and anti-MOG autoantibodies, respectively [16]. In other large cohort studies with recurrent ON with or without other demyelinating lesions conducted in China, South Korea, and Germany, anti-MOG antibodies were found in 6.3~18.3% of the participants [17,18,19]. (Appendix A). Overall, these studies highlight the comparable prevalence of MOGAD among various populations across the globe.

### 2.2. Characteristics of Visual Symptoms 

In patients with MOGAD, ON stands as the most frequent symptom. Those with anti-AQP4 autoantibodies experienced a significantly higher rate (53%) of severe disturbance in visual acuity compared to patients with anti-MOG autoantibodies. Patients having anti-MOG autoantibodies exhibited a higher frequency of optic disc swelling and pain related to eye movement than those with anti-AQP4 autoantibodies. More than 95% of patients with anti-MOG autoantibodies displayed complete visual field loss or central scotoma, while patients with anti-AQP4 autoantibodies demonstrated diverse visual field abnormalities, including altitudinal hemianopsia, nasal hemianopsia, and temporal hemianopsia (Table 1). Among the characteristics, bitemporal hemianopsia, homonymous hemianopsia [20], and binocular vision loss from chiasmal lesions were relatively common in patients with anti-AQP4 autoantibodies but rare in those experiencing ON due to other causes [21]. 

In patients with anti-MOG autoantibodies, the primarily affected area was the proximal part of the optic nerve, especially the anterior intra-orbital part, which constituted over half of the total intra-orbital optic nerve. Inflammation in ON with anti-AQP4 autoantibodies usually is not extended to this proximal optic nerve portion and extension to the proximal portion was a significant prognostic factor in this group [22]. 

Within our ON patient cohort, MRI scans revealed swollen optic nerves in 91% of those with anti-MOG autoantibodies and 82% of those with anti-AQP4 autoantibodies [15] (Table 1). In differentiating between the two disorders, the presence or absence of optic disc swelling is a crucial observation. Patients with MOGAD typically demonstrate substantial swelling in both the optic disc and nerve, accompanied by pronounced inflammation encompassing the orbital tissues surrounding the optic nerve. This inflammation-induced pain during eye movement is further intensified by the effects of the dural sheath encompassing the optic nerve, the sclera, and intra-orbital tissues that influence the extraocular muscles at the common tendinous ring through the trigeminal nerve. In patients with anti-AQP4 autoantibody-positive ON, lesions were also detected at sites outside of the optic nerve, including cerebral white matter, brainstem, and particularly spinal cord. In our study, lesions in the spinal cord were found in eight patients (22%), including one patient with longitudinally extensive transverse myelitis [15]. 

### 2.3. ON Recurrence

ON with anti-MOG autoantibodies exhibits a considerable recurrence rate (44–83%), similar to that observed in anti-AQP4 autoantibody-positive patients [23]. However, biomarkers that can predict ON relapses are lacking. Contentti et al. noted that patients with higher anti-MOG autoantibody titers or those with rapidly disappearing antibodies after treatment during the acute phase tended to show a monophasic clinical course. They recommended that treatment with oral steroids could be tapered and discontinued at six months following the initial relapse [24,25]. Within our cohort, 353 of 531 patients (66.5%) were experiencing relapsing ON episodes. Unfortunately, long-term follow-up data were not available, and the medications used for relapse prevention varied among the participating facilities. According to reports, the greatest risk of relapse occurs during the first year and remains high within the next 5 years [26,27]. 

## 3. Other Immunologic Parameters 

In addition to anti-AQP4 autoantibodies, various systemic autoantibodies are frequently detected in patients with NMOSD, including antinuclear, anti-Sjögren’s syndrome type-A and -B, antithyroid stimulating hormone receptor antibody, antithyroglobulin, and anti peroxidase autoantibodies [28]. However, these autoantibodies are not commonly found in patients with ON due to MOGAD. 

Anti-MOG autoantibodies primarily exist as IgG1 isotypes capable of activating the complement. However, the role of complement activation in MOGAD remains a focus of debate and is not well established. In contrast, complement activation has a strong association with ON pathogenesis in patients with anti-AQP4 autoantibodies, rendering treatment with complement inhibitors a potential approach. In a rodent model of ON, the administration of human anti-MOG-IgG in combination with human complement resulted in low levels of complement deposition [29]. The administration of anti-MOG antibodies that cross-reacted with rodent MOG led to increased T cell infiltration and complement deposition in addition to the observation of MOG- or myelin basic protein-specific T cells [29]. On the other hand, genetic studies in MOGAD show no strong correlation between human leukocyte antigen genotypes [30]. 

The cytokine and chemokine profiles of patients with MOGAD show high levels of T-helper (Th)17-related cytokines and chemokines [31]. These findings suggest that both Th17 and Th1 cells, along with B cells, may contribute to the pathogenesis of MOGAD. However, memory cells and long-lived plasma cells were also elevated in patients with MOGAD [32]. In one study, MOG-specific B cells did not correlate with the serum anti-MOG-IgG autoantibody titer [33]. Moreover, human anti-MOG-IgG autoantibodies were shown to induce natural killer cell-mediated death of MOG-expressing cells in vitro [34,35]. 

MOG-IgG was detected in the CSF of 12 of 18 (67%) patients with seropositivity for anti-MOG-IgG autoantibodies, suggesting an extrathecal origin. The oligoclonal IgG bands were not commonly detected in patients with MOGAD, while anti-MOG-IgG antibodies were present at disease onset and remained detectable in 40 out of 45 (89%) follow-up samples obtained over a median period of 16.5 months (range 0–123 months) [19].

## 4. Immunopathology of MOGAD

Although systematic neuropathological evaluation of MOGAD patients is rare, several studies including autopsy and biopsy samples from patients with anti-MOG autoantibodies have revealed a distinct pattern of perivenous and confluent demyelination in white matter, cortex, and deep gray matter structures [36,37,38].

In a study of biopsy samples, meningeal inflammation was observed in 86% of the cases, subpial lesions were present, and active demyelinating areas showed an abundance of myelin-laden macrophages/microglial cells [36]. However, the majority of the infiltrating lymphocytes were CD4-positive, with few B cells and CD8+ T cells [36,37].

In some studies, complement activation was demonstrated in active lesions, resembling pattern II demyelinating lesions of MS [36], but it was largely absent in another study of 11 biopsies [37]. Additionally, the destruction of oligodendrocytes displayed a varying pattern and selective MOG loss was not observed [36]. However, the loss of MOG expression was described in another study by Takai et al. [37] who showed that most of the demyelinating lesions exhibited a perivenous demyelinating or fusion pattern mainly in the corticomedullary junction and white matter, suggesting that ADEM-like perivenous inflammatory demyelination was a characteristic finding of MOGAD. The early-phase demyelinating lesions of MOGAD exhibited MOG-dominant myelin loss with relatively preserved oligodendrocytes. This feature distinguishes MOGAD from anti-AQP4 autoantibody-related NMOSD, including pronounced perivascular deposition of immunoglobulins and complement together with demyelinating lesions containing myelin degradation products in numerous macrophages [39]. The pathologic features of MOGAD are clearly different from those of MS and NMOSD, suggesting an independent autoimmune demyelinating disease entity [37]. The optic nerve is a vulnerable organ in MOGAD which might be based on that both the protein and mRNA expression levels of MOG are higher in the optic nerve than in the spinal cord and brain in mice [40,41].

## 5. Treatment and Visual Outcome

There are currently no randomized control trials or evidence-based guidelines for the treatment of acute disease and relapse in patients with MOGAD [42]. In the acute stage, most patients with MOGAD are treated with high-dose methylprednisolone pulse therapy with or without intravenous immunoglobulin therapy (IVIg) and plasmapheresis, with favorable response observed. Recovery was significantly better in patients with anti-MOG autoantibodies than in those with anti-AQP4 autoantibodies who require additional plasmapheresis [43]. In patients displaying resistance to these mentioned treatments, alternative therapeutic approaches, such as immunosuppressants (azathioprine, cyclophosphamide, tacrolimus, mycophenolate mofetil), satralizumab, or B cell depletion therapy, may be considered [25,44,45,46]. In the nationwide survey of epidemiological and clinical characteristics of Japanese patients with MOGAD, the favorable therapeutic effect of tacrolimus was shown as 72.7% (40 out of 55 treated with tacrolimus) [47]. 

While monthly intravenous immunoglobulin treatment was associated with a reduction in annual relapse rate in pediatric and adult cohorts, 20–71% of treated patients experienced relapses [48]. Moreover, some disease-modifying treatments used for MS, including fingolimod or natalizumab, might induce severe relapse in patients with MOGAD [49]. 

### 5.1. Recent Novel Therapeutics 

#### 5.1.1. IL-6 Receptor Inhibitor

IL-6 is a proinflammatory cytokine whose signaling pathway is triggered by complement deposition; IL-6 promotes B cell stimulation, blood–brain barrier dysfunction, leukocyte migration, and cytokine and chemokine production [50,51,52]. In one study, 73% (n = 11) of the patients with MOGAD treated with tocilizumab, a humanized IL-6 receptor inhibitor, for 12 months remained relapse-free, which was higher than the relapse-free rate of 57% (n = 28) observed in patients with anti-AQP4 autoantibody-related NMOSD [52].

#### 5.1.2. Rituximab

One of the most frequently used drugs in MOGAD is rituximab, which targets CD20+ B cells [53]. However, despite efficient B cell depletion, only 55% and 33% of the patients treated with rituximab were relapse-free in 1 and 2 years after treatment, respectively [54]. Thus, B cell depletion was less effective in patients with MOGAD than in those with anti-AQP4 autoantibody-related NMOSD, indicating that B cells are not the only effector in MOGAD. 

#### 5.1.3. Inebilizumab

Regarding inebilizumab, a humanized anti-CD19 monoclonal antibody, six of seven patients with anti-MOG autoantibody positivity did not experience relapse during the follow-up period of 210 days [55]. Inebilizumab was generally well tolerated and the adverse event profile observed was similar to that of anti-AQP4-positive patients.

## 6. Survey of Antibody-Binding Sites of Anti-MOG Autoantibodies in Patients with MOGAD

Conformation-dependent MOG-specific antibodies can initiate demyelination in EAE, as demonstrated by the evaluation of anti-MOG antibodies in MS [56], even if denatured MOG protein could still trigger T cell immunity.

MOG is essential for the formation, maintenance, and degradation of the myelin sheath through its adhesive properties and by mediating interactions between myelin and the immune system. The structure of the extracellular N-terminal portion of MOG forms an immunoglobulin variable (Ig-V) fold consisting of two antiparallel β-sheets located in the FG loop; this fold is recognized by the rodent anti-MOG monoclonal antibody 8-18C5 [57,58,59,60]. Up to 15 MOG splice variants have been described in humans and non-human primates. Anti-MOG-IgG has been shown to bind to six major MOG isoforms. The human MOG epitopes most frequently recognized by anti-MOG autoantibodies are located within the extracellular Ig-V-like domain. In this region, proline 42, located in the CC’ loop, is the most important amino acid for antibody recognition, followed by histidine 103 and serine 104, both located in the FG loop (Figure 1). In addition, all monoclonal antibodies bind to the native glycosylated extracellular domain of MOG expressed at the cell surface, and six of them recognize pure discontinuous epitopes [61]. 

We have previously explored the binding site of anti-AQP4 autoantibodies by exploiting the differences in AQP4 structure between humans, mice, and rats [62,63]. Using these strategies, we examined the binding site of MOG antibodies with three extracellular N-terminal exons exchanged between human and mouse MOG and found that exon 2 of human MOG was the major binding site for anti-MOG autoantibodies in patients with ON due to MOGAD (presented at ECTRIMS 2018 Scientific Session 15 Berlin 12 October 2018).

Mayer et al. investigated the binding epitopes of anti-MOG autoantibodies in the sera of 111 patients, including 104 children and 7 adults, with anti-MOG autoantibodies, including patients with ADEM, TM, ON, MS, NMOSD, and chronic relapsing inflammatory ON [64]. A human mature MOG is a protein containing a single peptide of 29 amino acids followed by 218 amino acids of the mature protein. The authors constructed several expression vectors harboring different mutations in MOG peptides and tested the binding ability of patient serum samples to these recombinant human MOG proteins (hMOG) (N31D, S104E, H103A/S104E, P42S, P42S/H103A/S104E, R9G/H10Y, and R86Q) in comparison to the wild-type human and mouse MOG proteins (mMOG). In 52 of 111 patients, the anti-MOG autoantibody response was directed against a single epitope in proline 42. In addition, the tip of the FG loop of MOG was detected in the serum samples of 36 patients (32%). However, the positions of P42S and H103A/S104E were detected in 19 out of 32 samples. Using the combination of each mutant, the authors showed seven different patterns of antibody binding in patients with MOGAD. However, the most frequently recognized epitopes were found in the CC’ and FG loops of hMOG. In addition, reactivity to both loops is the most common combination among sera recognizing multiple epitopes that are too far apart to be recognized together with an antibody to a single binding epitope [65], suggesting that patients’ sera contain at least two distinct antibody populations. Overall, half of the patients showed an immune response directed against a single epitope, whereas the remaining patients showed recognition of multiple epitopes. The observed pattern of recognition was not related to the clinical phenotypes and this recognition pattern remained constant over the observation period of 50 months. 

Bettelli et al. [41] generated MOG-specific T cell receptor (TCR) transgenic mice of the C57BL/6 strain expressing a TCR composed of Vα3.2 and Vβ11. These mice developed ON without involvement of other neuronal tissues. And these mice preferentially showed ON when immunized with MOG_35-55_ peptide and MOG protein without showing encephalitis or myelitis. When the TCR encounters an antigen–MHC complex that fits the binding site, it initiates a signaling cascade inside the T cells. Upon antigen recognition, the T cells are activated and proliferate. The mice harboring the TCR that preferentially recognizes antigen expressed on optic nerves tend to develop ON without accompanying encephalitis or myelitis, which might suggest that MOG in the optic nerve has a specific binding epitope different from that in other nervous tissues or the initial target of MOGAD might be the optic nerve. In particular, the optic nerve is reported to show higher expression of MOG compared to the spinal cord. However, it has not been shown that the anti-MOG autoantibodies themselves produced in these mice specifically recognize the optic nerve. It is also not clear that the structure of the epitopes is associated with this antibody binding. Although there are many unanswered questions, this is one of the suggestions for the relationship between antigen epitope and clinical phenotypes. In humans, autoantibody production is usually polyclonal and has multiple binding capabilities, so another factor is needed to evaluate the relationships between antigen epitopes and clinical phenotypes.

Autoantibodies produced in peripheral lymphoid organs must traverse the blood–brain barrier to encounter antigens in the CNS. MOG-IgG from the acute phase of MOGAD patients has been shown to activate endothelial cells that form the blood–brain barrier with increased vascular cell adhesion molecule (VCAM)-1 and intracellular adhesion molecule (ICAM)-1 proteins [65,66]. Autoantibodies other than MOG-IgG may also directly or indirectly disrupt the functions of adhesion molecules. It is also possible that there are differences in the nature of blood–tissue barriers between the optic nerve and the CNS. 

Based on several lines of research, it is suggested that the sera from MOGAD patients initially target the blood–tissue barrier of the optic nerve, causing edematous swelling of the optic nerve before directly binding anti-MOG autoantibodies to the myelin sheath.

## 7. Conclusions

The clinical spectrum of MOGAD is expanding. The most common clinical phenotype in children is ADEM with or without ON, whereas adult patients often develop ON and myelitis or cerebral cortical encephalitis. Compared to patients with anti-AQP4 autoantibody-associated NMOSD, patients with MOGAD respond well to immunotherapy; however, some patients experience multiple relapses. Detection of anti-MOG autoantibodies requires a cell-based assay using structure-preserved antigens. However, most of the serum samples that recognized human MOG did not recognize mouse MOG well, highlighting that the diagnosis may be missed if the presence of anti-MOG autoantibodies is evaluated only with rodent tissue. 

Factors associated with age-specific clinical phenotypes with lesion selectivity in the CNS in MOGAD patients are unknown. In addition, the functions of MOG protein are not fully understood and the pathogenicity of anti-MOG autoantibodies in MOGAD has not been demonstrated. To study the role of anti-MOG autoantibodies, it is necessary to develop a model system; however, human anti-MOG autoantibodies have low affinity for mouse MOG, which interrupts the development of the disease model.

The anti-MOG autoantibody-binding sites on MOG in the CNS have been extensively studied and it has been found that the most important part for human antibody recognition is located in the CC’ loop with proline at amino acid number 42, followed by histidine 103 and serine 104 in the FG loop, but the antibody-binding epitopes differ between patients without clear relevance to the clinical phenotype. It is possible that triggering factors other than anti-MOG autoantibodies are involved in MOGAD target vessel walls, blood–brain barrier structures, etc. to form barrier–loose edematous lesions. Among the patients affected by the site-directed pathogenicity of anti-MOG autoantibody-mediated inflammation, only a small number of studies have examined the different clinical phenotypes of MOGAD, and a better understanding of the underlying immunopathology requires the evaluation of more patients with different clinical phenotypes in the future.

## Figures and Tables

**Figure 1 ijms-24-13368-f001:**
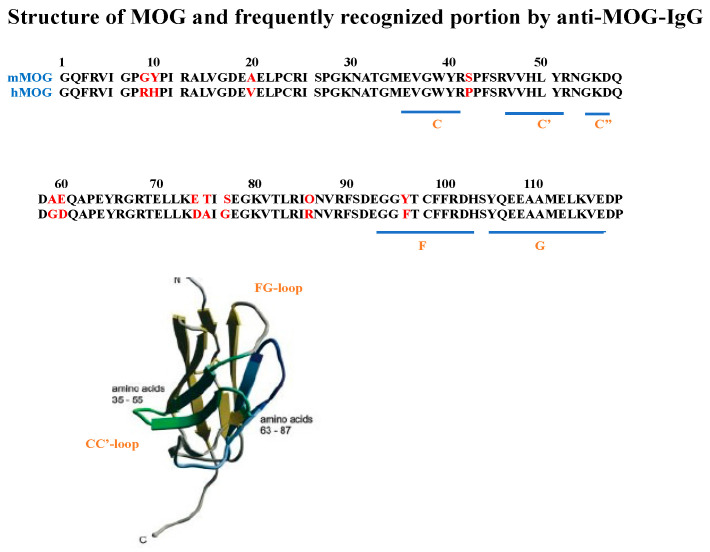
Anti-MOG-IgG autoantibody-binding sites on the extracellular portion of MOG identified in patients with MOGAD.

**Table 1 ijms-24-13368-t001:** Brief summary of the optic neuritis patients in a large Japanese cohort (Ishikawa H, Ophthalmology 2019).

Total	MOG (+)	AQP4 (+)	Both (−)
531 cases	(54 cases: 10.2%)	(66 cases: 12.4%)	(410 cases: 77.2%)
Female ratio	51.0%	84.1%	63.7%
Age at onset	42.9 ± 19.3	51.1 ± 14.0	48.0 ± 10.0
Annual relapse rate	1.56 ± 0.7	1.50 ± 0.4	0.6 ± 0.4
Nadir visual acuity	0.15	0.09	0.1
Outcome of visual	0.93	0.54	
acuity after 1st Tx.			
Pre Tx. 0.3>	76.8%	82.5%	73.8%
Post Tx. 0.7<	80.0%	44.9%	56.8%
Optic disc swelling	34/45 (76%)	21/61 (34%)	166/361 (46%)
Pain on eye movement	36/47 (77%)	31/59 (53%)	161/347 (46%)
Visual field loss			
Complete	9/41 (22%)	14/55 (26%)	45/311 (15%)
central/temporal/altitude/nasal (%)	46/7/22/0	73/0/2/2	61/4/15/5
MRI			
Optic nerve swelling	91%	82%	67%
Optic nerve lesion			
(ant./post./entire/chiasma) (%)	44/22/34/8	24/49/27/10	41/44/15/6
Lesion length (long/short) (%)	61/39	67/33	47/53

## Data Availability

The data presented in the Table 1 is openly available in reference [15].

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
