# Peer review of "Pathogenesis, Clinical Features, and Treatment of Patients with Myelin Oligodendrocyte Glycoprotein (MOG) Autoantibody-Associated Disorders Focusing on Optic Neuritis with Consideration of Autoantibody-Binding Sites: A Review"

_ijms, 2023, doi:10.3390/ijms241713368_

Round 1

Reviewer 1 Report

My main criticism of the paper is that this seems to be using an already published article (Ishikawa et al, Ophthalmology 2019;126 (10): 1385) without any additional data, and then repurposed as a review. Even when reference to other articles are made (sections 3 and 4), it does not appear to be cohesive and feels like a mix and match of information from other MOGAD articles. This “lack of flow” is also shown in the treatment section when there is passive mention about Inebilizumab in the N-Momentum study and coupling it with a drug which has a completely different action (Eculizumab). The title suggests autoantibody binding sites, but the data is not shown when this is again casually mentioned (line 260).

It feels like this is old wine in a new bottle, which is not giving the readers any new information. The authors’ original article (Ophthalmology 2019) gives more or less the same (if not more) information as this one and am not entirely sure whether yet another article mentioning what is already well known is relevant. (I can summarise the manuscript in two lines - i.e. ON with swelling and pain on eye movements are very common in MOGAD, and is often severe but responsive to therapies. Cell-based assays are useful, but because there is low affinity to the mouse MOG molecule, more studies are needed with innovative models).

Minor comments:

Unable to see any Tables/Figures (but irrelevant given the comments above and also presumably they are not dissimilar to the ones published in 2019).

Line 44 – rephrase sentence without repeating “binding”

Line 118 – characteristic of visual symptoms – might be better in a comparative table (Aqp, MOG, double-negative), rather than long text –however, this is how it is described in the original paper (Ophthalmology 2019).

Line 136 – spellcheck – “annti..”

Lines 122 vs 138 – appreciate that one may be direct visualisation and the other MRI, but would avoid these as two paragraphs, since there is unnecessary repetition – best to amalgamate all into the table above – may be sub-dividing (optic disc swelling 76% (fundoscopy); 91% (MRI) etc)

Line 147 – remove repeated “lesions”

Line 192 – better to say “confluent perivenous demyelination”?

Nothing major - few minor typos and grammatical errors

Author Response

Reply to Reviewer 1

My main criticism of the paper is that this seems to be using an already published article (Ishikawa et al, Ophthalmology 2019;126 (10): 1385) without any additional data, and then repurposed as a review. Even when reference to other articles are made (sections 3 and 4), it does not appear to be cohesive and feels like a mix and match of information from other MOGAD articles.

→We need to accept the criticism with honesty. The paper published by Ishikawa et al. was prepared by all ophthalmologists except for me. Most of the recent review articles on MOGAD were from the neurology field and focused on comparing AQP4 seronegative NMOSD and the unique phenotype of encephalitis. This led us to examine the phenotype of optic neuritis from a neurology perspective. However, there have been similar review papers published quickly. In order to express our stance, we included the sentence “The frequency of optic neuritis suggests that the optic nerve is the most vulnerable lesion in MOGAD. During the acute stage, the optic nerve shows significant swelling with severe visual symptoms, and an MRI of the optic nerve and brain lesion tends to show an edematous appearance. These features can be alleviated with early extensive immune therapy, which may suggest that the initial attack of anti-MOG autoantibodies could target the structures on the blood-brain barrier or vessel membrane before reaching MOG protein on myelin or oligodendrocytes.” in the Abstract section.

This “lack of flow” is also shown in the treatment section when there is passive mention about Inebilizumab in the N-Momentum study and coupling it with a drug which has a completely different action (Eculizumab).

→The study about Inebilizumabwas originally targeted towards AQP4-positive NMOSD, but we found that six patients with AQP4(-)/MOG(+) in that study actually showed better response to the inebilizumab administration. We replaced the previous reference with one that discusses the effects of the treatment on MOG(+) patients instead.

The title suggests autoantibody binding sites, but the data is not shown when this is again casually mentioned (line 260).

→Our study on anti-MOG binding in relation with clinical phenotype have not been published yet and could not show new data. We referred our abstract of oral presentation at ECTRIMS 2019 in page 10 to 11.

It feels like this is old wine in a new bottle, which is not giving the readers any new information. The authors’ original article (Ophthalmology 2019) gives more or less the same (if not more) information as this one and am not entirely sure whether yet another article mentioning what is already well known is relevant. (I can summarise the manuscript in two lines - i.e. ON with swelling and pain on eye movements are very common in MOGAD, and is often severe but responsive to therapies. Cell-based assays are useful, but because there is low affinity to the mouse MOG molecule, more studies are needed with innovative models).

→We agree for the reviewer’s comment. We moved some redundant description into Table 1. and shortened 2.2 Characteristics of visual symptoms part in the text.

Minor comments:

Unable to see any Tables/Figures (but irrelevant given the comments above and also presumably they are not dissimilar to the ones published in 2019).

→We updated previous Table 1 to that contained the same information as previously shown.

Line 44 – rephrase sentence without repeating “binding”

→We replaced repeatedly used “binding” to other words as “affinity”.

Line 118 – characteristic of visual symptoms – might be better in a comparative table (Aqp, MOG, double-negative), rather than long text –however, this is how it is described in the original paper (Ophthalmology 2019).

→Thank you for your comments. I have updated Table 1 to reflect the new information.

Line 136 – spellcheck – “annti..”

→corrected as “anti- “

Lines 122 vs 138 – appreciate that one may be direct visualisation and the other MRI, but would avoid these as two paragraphs, since there is unnecessary repetition – best to amalgamate all into the table above – may be sub-dividing (optic disc swelling 76% (fundoscopy); 91% (MRI) etc)

→We moved the MRI findings into Table 1.

Line 147 – remove repeated “lesions”

→We removed repeated “lesions”

Line 192 – better to say “confluent perivenous demyelination”?

→We corrected this expression as “confluent demyelination in white matter, cortex, and deep gray matter structures”

Please find the replies in the document attached.

Reviewer 2 Report

The work analyzes in detail the characteristics of a rare neuroimmunological disease, MOG-associated demyelination. It covers epidemiological, clinical characteristics and treatment. It touches on the pathomechanism, and also covers the difficulties of researching the topic. The topic is important and the thesis can help general neurologists, neuroimmunologists, clinicians and researchers. What is the advantage is also the disadvantage to some extent - the article talks about the topic very widely, which makes it a very concise and difficult read in some places. The comparison with NMOSD runs throughout the composition, which seems to be a useful theme.

Overall, we are talking about a precise and detailed, well-written thesis, which, in my opinion, is suitable for publication.

Rarely used expressions sometimes appear (albeit, akin to), but this is not disturbing. In the 147th line there is word repetition (lesions).

Author Response

Reply to Reviewer 2.

The work analyzes in detail the characteristics of a rare neuroimmunological disease, MOG-associated demyelination. It covers epidemiological, clinical characteristics and treatment. It touches on the pathomechanism, and also covers the difficulties of researching the topic. The topic is important and the thesis can help general neurologists, neuroimmunologists, clinicians and researchers. What is the advantage is also the disadvantage to some extent - the article talks about the topic very widely, which makes it a very concise and difficult read in some places. The comparison with NMOSD runs throughout the composition, which seems to be a useful theme.

→ We would greatly appreciate for your constructive and thoughtful comments. We checked redundant expressions and attempted to organize the information in the table, and novel therapies into separate headings.

Rarely used expressions sometimes appear (albeit, akin to), but this is not disturbing. In the 147th line there is word repetition (lesions).

→Thank you for your suggestions. Following your comments, we replaced “albeit” to “although” and “akin to” to “resembling”.

Please find the responses in the document attached.
